# QUARK: A GRADIENT-FREE QUANTUM LEARNING FRAMEWORK FOR CLASSIFICATION TASKS

## ABSTRACT

As more practical and scalable quantum computers emerge, much attention has been focused on realizing quantum supremacy in machine learning. Existing quantum ML methods either (1) embed a *classical* model into a target Hamiltonian to enable quantum optimization or (2) represent a quantum model using variational quantum circuits and apply *classical* gradient-based optimization. The former method leverages the power of quantum optimization but only supports simple ML models, while the latter provides flexibility in model design but relies on gradient calculation, resulting in barren plateau (i.e., gradient vanishing) and frequent classical-quantum interactions. To address the limitations of existing quantum ML methods, we introduce Quark, a gradient-free quantum learning framework that optimizes *quantum* ML models using *quantum* optimization. Quark does not rely on gradient computation and therefore avoids barren plateau and frequent classical-quantum interactions. In addition, Quark can support more general ML models than prior quantum ML methods and achieves a dataset-size-independent optimization complexity. Theoretically, we prove that Quark can outperform classical gradient-based methods by reducing model query complexity for highly non-convex problems; empirically, evaluations on the Edge Detection and Tiny-MNIST tasks show that Quark can support complex ML models and significantly reduce the number of measurements needed for discovering near-optimal weights for these tasks.

## 1 INTRODUCTION

Quantum computing provides a new computational paradigm to achieve exponential speedups over classical counterparts for various tasks, such as cryptography (Shor, 1994), scientific simulation (Tazhigulov et al., 2022), and data analytics (Arute et al., 2019). A key advantage of quantum computing is its ability to entangle multiple quantum bits, called *qubits*, allowing $n$ qubits to encode a $2^n$-dimensional vector, while encoding this vector in classical computing requires $2^n$ bits.

Inspired by this potential, recent work (Jaderberg et al., 2022; Macaluso et al., 2020b; Torta et al., 2021; Kapoor et al., 2016; Bauer et al., 2020; Farhi & Neven, 2018a; Schuld et al., 2014; Cong et al., 2019b) has focused on realizing quantum speedups over classical algorithms in the field of supervised learning. Existing quantum ML work can be divided into two categories: *classical model with quantum optimization* (CMQO) and *quantum model with classical optimization* (QMCO). First, CMQO methods embed a classical ML model jointly with the optimization problem into a target Hamiltonian and optimize the model using *quantum adiabatic evolution* (QAE) (Finnila et al., 1994) or *quantum approximate optimization algorithm* (QAOA) (Farhi et al., 2014; Torta et al., 2021). As the transition between a classical model and the target Hamiltonian only applies to low-order polynomial activations (see Figure 2), CMQO methods do not support ML models with non-linear activations that cannot be represented in low-order polynomial (e.g., ReLU). Second, QMCO methods optimize variational quantum models [1] by iteratively performing gradient descent using classical optimizers. QMCO methods are fundamentally limited by barren plateau (i.e., gradient vanishing (McClean et al., 2018)) and the high cost of frequent quantum-classical interactions.

To address the limitations of existing quantum ML methods, we introduce Quark, a gradient-free quantum learning framework for classification tasks that optimizes *quantum models with quantum*

---

[1]They are also known as variational quantum circuits (VQC)-based models in the quantum literature.

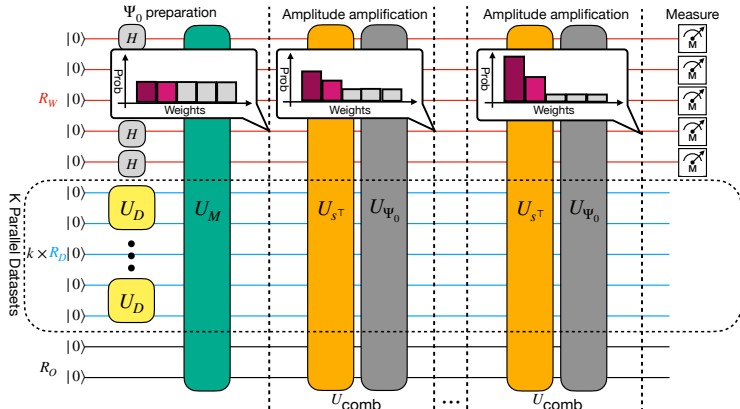

Figure 1: An overview of the Quark optimization framework. Each horizontal line indicates a qubit, and each box on these lines represents one or multiple quantum gates applied on these qubits. Quark's optimization pipeline includes three stages: (1) $\Psi_0$ preparation, which initializes $R_W, R_D, R_O$ and performs model's forward processing $U_M$, (2) amplitude amplification using a Grover-based algorithm, and (3) weights measurement. Quark uses $K$-parallel datasets (KPD) to maximize the probability of observing highly accurate weights in each measurement. Darker bars in the probability plots denote weights with higher accuracies.

*optimization* (QMQO). Figure 1 shows an overview of Quark. A key idea behind Quark is entangling the weight [2] of an ML model (i.e., the $R_W$ register in Figure 1) and the encoded dataset (i.e., the $R_D$ register in Figure 1) in a quantum state, where model weights that achieve optimal classification accuracy on the training dataset can be observed with the highest probabilities in a measurement. Therefore, users can obtain highly accurate model weights by directly measuring the updated $R_W$ weight register. To maximize the probability of observing optimal weights, we introduce two key techniques.

**Amplitude amplification.** Quark uses a Grover-based mechanism to *iteratively* update the probability distribution of weights based on their training accuracies. As a result, the probability of observing weights with higher accuracy increases after each Grover iteration, as shown in Figure 1.

$K$**-parallel datasets (KPD).** Applying amplitude amplification on one dataset results in a linear amplification scenario where the measuring probability of each weight is proportional to its training accuracy $J(w_i)$. We further introduce $K$-*parallel datasets*, a technique to enable *exponential* amplification. Specifically, by entangling $k$ identical training datasets with model weights in parallel (using $k \times R_D$, as shown in Figure 1), the probability of observing weight $w_i$ in a measurement is proportional to $J(w_i)^k$. Therefore, as $k$ increases, the optimized probability distribution of weights gradually converges to the optimal weights.

Compared with CMQO methods, Quark provides more flexibility in model design by composing models directly on quantum circuits and therefore supports a broader range of ML models. Compared with QMCO methods, Quark does not require gradient calculation and therefore does not suffer from barren plateau. Quark avoids frequent classical-quantum interactions by realizing both model design and optimization fully on quantum. Besides, by using basis encoding for the training dataset, Quark supports non-linear operations (e.g., ReLU) in its model architecture, and the optimization complexity is independent of the training dataset size.

Theoretically, we compare model query complexity[3] between Quark and gradient-based methods on a balanced $C$-way classification task, and prove that Quark can outperform gradient-based methods by reducing model query complexity for highly non-convex problems. In addition, we prove that using $K$-parallel datasets can further reduce model query complexity under certain circumstances.

Simulations on two tasks (i.e., Edge Detection and Tiny-MNIST) show that Quark supports complex ML models, which can include quantum convolution, pooling, fully connected, and ReLU layers.

---

[2]Throughout the paper, we use the term *weight* to refer to the set of *all* trainable parameters of a model.

[3]Number of model forward being called

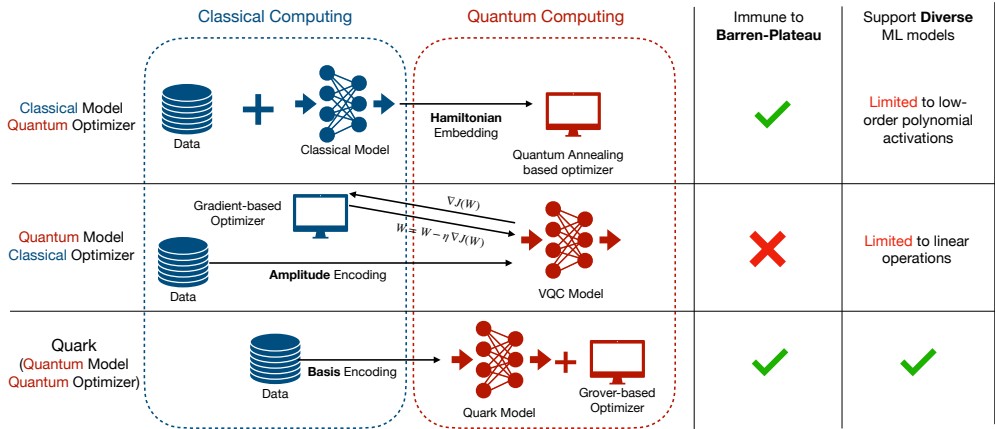

Figure 2: Comparison between CMQO, QMCO, and Quark (QMQO).

In addition, Quark can significantly reduce the number of measurements needed for discovering a near-optimal weight by applying amplitude amplification and KPD.

**Contributions.**  This paper makes the following contributions:

- We propose Quark, a gradient-free quantum learning framework for classification tasks that optimizes quantum ML models with quantum optimization. Quark avoids barren plateau and frequent classical-quantum interactions, supports more general models than prior quantum ML frameworks, and achieves a dataset-size-independent optimization complexity.

- Theoretically, we prove that Quark can outperform gradient-based methods by reducing model query complexity for highly non-convex problems and that using KPD can further reduce model query complexity.

- Empirically, we show that Quark can support complex ML models and significantly reduce the number of measurements needed for discovering a near-optimal weight for the Edge Detection and Tiny-MNIST tasks.

## 2  RELATED WORK

Figure 2 compares Quark with existing quantum ML approaches.

### 2.1  CLASSICAL MODEL WITH QUANTUM OPTIMIZATION

Existing CMQO methods aim at solving classical ML problems with quantum optimization techniques by leveraging the advantage of quantum parallelism (Nielsen & Chuang, 2002). Based on the well-established algorithmic foundation in quantum annealing (Finnila et al., 1994; Kadowaki & Nishimori, 1998; Brooke et al., 1999; Santoro et al., 2002; Santoro & Tosatti, 2006) and adiabatic quantum computing (Farhi et al., 2001; Albash & Lidar, 2018), prior work (Denil & De Freitas, 2011; Dumoulin et al., 2014; Adachi & Henderson, 2015) attempts for the quantum restricted Boltzmann machine (RBM) by formulating RBM as an Ising model (Cipra, 1987). Inspired by the quantum approximate optimization algorithm (QAOA) (Farhi et al., 2014), Torta et al. (2021) embeds a single binary perceptron layer into a target Hamiltonian to search for optimal weights. However, CMQO methods are limited by the locality restriction of the target Hamiltonian and can only embed models with low-order polynomial activations (e.g., square). This limitation prevents CMQO methods from supporting practical deep learning architectures, which generally contain non-polynomial activations such as ReLU and sigmoid.

Similar to Quark, Kapoor et al. (2016) also uses Grover's algorithm to find a hyperplane that can perfectly separate the training dataset. However, this method only applies to an idealistic setup where a hyperplane with perfect classification exists in its search space. Besides, the method cannot adapt to generic model architectures other than single-layer perceptrons.

## 2.2 QUANTUM MODEL WITH CLASSICAL OPTIMIZATION

Motivated by the recent advances in variational quantum algorithms (VQAs) (Cerezo et al., 2021), QMCO methods use variational quantum circuits (VQC) (Benedetti et al., 2019) to represent the trainable parameters of an ML model. Havlíček et al. (2019); Schuld & Killoran (2019) use VQC as a variational feature map to reproduce linear support vector machines (SVM) and kernel methods on quantum circuits, which can outperform classical counterparts under certain circumstances (Liu et al., 2021).

Besides conventional ML methods, recent work has also explored the feasibility of classical neural networks on quantum circuits (Massoli et al., 2022). Farhi & Neven (2018b); Macaluso et al. (2020a); Killoran et al. (2019) use VQC as building blocks for their quantum perceptron models with a classical gradient-based optimizer. Quantum dissipative neural network (Beer et al., 2020) (QDNN) and quantum convolutional neural network (Cong et al., 2019a) (QCNN), on the other hand, move a step forward towards more complicated neural architectures. QDNN enlarges its model space by applying unitary operators on both the input and output qubits, while QCNN uses a measurement-controlled operation to enable non-linear operations. However, McClean et al. (2018) shows that the *barren plateau* phenomenon commonly exists in VQC-based methods, where gradients vanish exponentially with the model size. Though Beer et al. (2020) claims to design a VQC model immune to barren plateau, Sharma et al. (2022) contradicts such claim with analytical proof. Though Du et al. (2021) uses a Grover algorithm as part of their method, they still require VQC as their model building blocks that require gradients update.

Besides, due to amplitude-based data encoding, VQC-based methods in general suffer from a linear dependency with respect to dataset size in terms of model query complexity during training. Another drawback for amplitude encoding is that due to the unitary constraint of quantum transformations, non-linear operations are hard to implement for VQC-based methods. In contrast, our method uses basis encoding that concerns only qubits state transformation rather than amplitude transformation, which enables more efficient model query complexity and more general non-linear transformations.

## 3 PRELIMINARIES

### 3.1 NOTATIONS

Let $\mathcal{D} = \{(x_i, y_i)\}_{i \in N}$ denote a training dataset, where $x_i \in \{0, 1\}^{d_x}$ is the binarized feature vector associated with the $i$-th sample, and $y_i \in \{0, 1\}^{d_y}$ is its label. Let $\hat{y} = f(w, \hat{x})$ : $\{0, 1\}^{d_w} \times \{0, 1\}^{d_x} \rightarrow \{0, 1\}^{d_y}$ denote our model parameterized by $w \in \{0, 1\}^{d_w}$. Given an objective $l(\hat{y}, y)$, our goal is to find a near-optimal $w^*$ that minimizes/maximizes the overall objective $J(w) = \frac{1}{N} \sum_{(x_i, y_i) \in \mathcal{D}} l(f(w, x_i), y_i)$. We focus on classification tasks and use the objective $l(\hat{y}, y) = \mathbb{1}(\hat{y} = y)$. We use $|\cdot|$ to denote the cardinality of a set and absolute value of a scalar, and use $\|\cdot\|_2$ and $\|\cdot\|_\infty$ to denote the $L_2$-norm and infinity norm of a vector. Finally, we use $\otimes$ to denote tensor product, and use $\neg, \oplus, \wedge,$ and $\vee$ to denote NEGATE, XOR, AND, and OR in logical expressions, respectively.

### 3.2 QUANTUM BASICS

A bit in the quantum regime, called a *qubit*, is represented by a super-position of $|0\rangle$ and $|1\rangle$, which is formally defined as $|z\rangle = \alpha|0\rangle + e^{i\phi}\beta|1\rangle$, where $\alpha$ and $\beta$ are the amplitudes, and $e^{i\phi}$ is the relative phase. Furthermore, the rule also enforces $\langle z|z \rangle = 1$ where $\langle z|$ is the conjugate transpose of $|z\rangle$. In an n-qubit system, a quantum state is represented as a superposition of $2^n$ basis states. For computational simplicity, we will be using the basis states that are spanned by $\{|0\rangle, |1\rangle\}^n$ and denote the superposition of a n-qubits state as $|\Psi\rangle = \sum_{i=0}^{2^n - 1} \frac{1}{\sqrt{2^n}}|i\rangle$ where $|i\rangle$ is the corresponding computational basis. We thus use $|w_i\rangle \in \{|0\rangle, |1\rangle\}^{d_w}$ to denote weight basis state, $|x_j, y_j\rangle$ for the entangled data basis state where $|x_j\rangle \in \{|0\rangle, |1\rangle\}^{d_x}$ is the feature and $|y_j\rangle \in \{|0\rangle, |1\rangle\}^{d_y}$ the label.

As a comparison, in the classical computing regime, each operation can only be applied to one state.

## 4 METHODOLOGY

To realize quantum supremacy, we circumvent the overhead induced by frequent classical-quantum interactions and the gradient calculation step in the VQC-based methods, which may result in barren

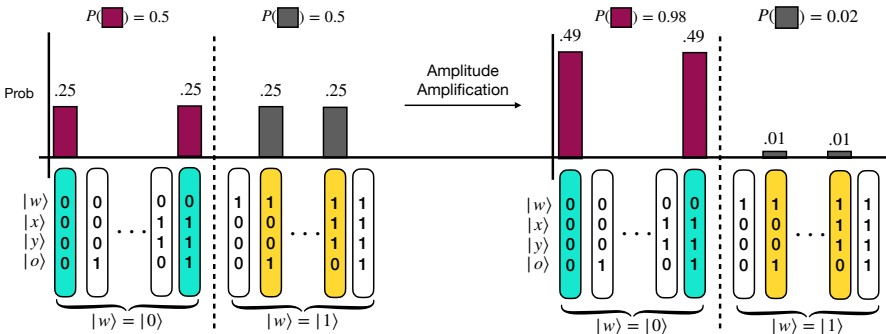

Figure 3: Toy illustration of Quark's insight, where the dataset $\{(x_0 = 0, y_0 = 0), (x_1 = 1, y_1 = 1)\}$ is constructed by oracle function $g(x) = 0 \oplus x$. We define our learner model as $o = f(x, w) = w \oplus x$. The colored qubit strings stands for the states activated by model forward (cyan for $|w\rangle = 0$ and yellow for $|w\rangle = 1$). violet bars stand for solution states measuring probability, gray stands for non-solution states measuring probability.

plateau as the circuit depth increases. In addition, as VQC-based methods use the amplitude encoding scheme for data encoding, non-linear operations are hard to implement, while their training time can linearly depend on the training dataset size.

To this end, we designed Quark in a QMQO fashion, which (1) achieves gradients-free optimization over the entire weight distribution, (2) uses basis encoding to achieve potential speedup as sample size scales up, and (3) enables a flexible quantum model design that can easily incorporate non-linearities. Figure 1 shows an overview of the quantum circuit design in Quark.

As we are using basis encoding over the entire data distribution, we initialize the data register $R_D$ as $|D\rangle = \sum_{(x_i, y_i) \in \mathcal{D}} \frac{1}{\sqrt{|\mathcal{D}|}} |x_i, y_i\rangle$. To initialize a model's weights register $R_W$, we construct a uniform state by applying the Hadamard gate on each weight qubit: $|W\rangle = \sum_{i=0}^{2^{d_w}-1} \frac{1}{\sqrt{2^{d_w}}} |w_i\rangle$. In addition, Quark includes an *auxiliary register* $R_O$ for storing intermediate results and model's output, which is initialized as $|O\rangle = |0\rangle^{d_o}$. Therefore, the initial state is represented as $|W\rangle \otimes |D\rangle \otimes |O\rangle$. By encoding the quantum model architecture as a unitary matrix $U_M$, the model's forward processing is defined as:

$$U_M \left( |W\rangle \otimes |D\rangle \otimes |O\rangle \right) = \sum_i \frac{1}{2^{\frac{d_w}{2}}} |w_i\rangle \sum_j \frac{1}{\sqrt{|\mathcal{D}|}} |x_j, y_j\rangle |f(w_i, x_j)\rangle \tag{1}$$

where $f(w_i, x_j)$ is the model's output for weight $w_i$ and sample $x_j$ [4]. Given the above expression for the model forward processing, the key insight for Quark is to update the probability of observing weight $w$ in a measurement based on its training objective $J(w)$. This is achieved through Grover's algorithm by defining the solution state space $\mathcal{S} = \{|w_i\rangle |x_j, y_j\rangle |f(w_i, x_j)\rangle \mid y_j = f(w_i, x_j)\}$. Appendix A.1 includes an introduction to the original Grover's algorithm. After amplitude amplification, we observe a model weight by measuring the weight register $R_W$.

To further illustrate our insight, Figure 3 shows a toy example, where the underline oracle function that generates the training data is $g(x) = 0 \oplus x$. Using one qubit for $x$, the training dataset is constructed as $\{(x_0 = 0, y_0 = 0), (x_1 = 1, y_1 = 1)\}$. By defining the learner model as $o = f(x, w) = w \oplus x$, where $w$ is the trainable parameter, Quark includes four qubits (i.e., $|w\rangle, |x\rangle, |y\rangle$, and $|o\rangle$). For initial state preparation, we simply go through the model forward circuit with a uniform weight initialization that result in an uniform amplitude state, as shown on the left of Figure 3. As no optimization happens at this stage, the probabilities for observing $|w\rangle = |0\rangle$ and $|w\rangle = |1\rangle$ in a measurement are equivalent. After amplitude amplification, Quark reaches an optimized amplitude state (shown on the right of Figure 3), where the probability of observing the optimal weight $|w\rangle = |0\rangle$ in a measurement is much higher.

For the rest of this section, Section 4.1 introduces Quark's on-circuit model design philosophy, Section 4.2 describes the Grover-based method for amplitude amplification, and Section 4.3 introduce

---

[4]We omit the model's intermediate results $|O\rangle$ for simplicity.

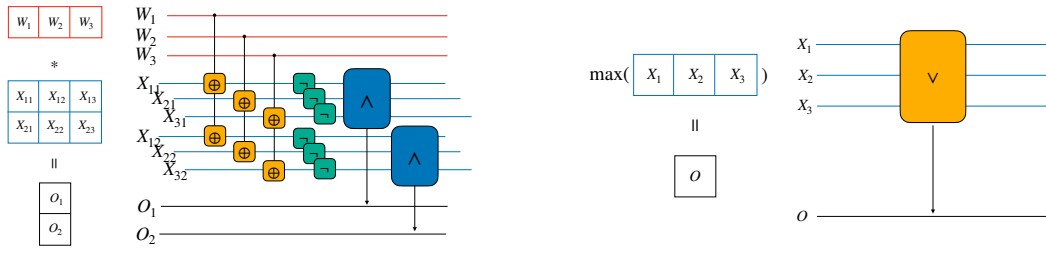

(a) Convolution            (b) MaxPool

Figure 4: Logical gates illustration of quantum module design. (a) **Convolution**: for a $1 \times 3$ Conv kernel with weights $W = [W_1, W_2, W_3]$ and input $X_i = [X_{i1}, X_{i2}, X_{i3}]$, the output is given by $O_i = (\neg(W_1 \oplus X_{i1})) \wedge (\neg(W_2 \oplus X_{i2})) \wedge (\neg(W_3 \oplus X_{i3}))$. (b) **MaxPooling**: for a input $X_i = [X_{i1}, X_{i2}, X_{i3}]$ the max pooling output is given by $O_i = X_{i1} \vee X_{i2} \vee X_{i3}$.

K-Parallel Dataset (KPD), a technique that enables exponential amplification to further improve the optimization algorithm.

### 4.1 MODEL DESIGN

The model circuit $U_M$ can be designed through arbitrary combinations of base gates (CNOT, Rotational gates, Hadamard gate, etc.), which can then entangle $|W\rangle, |D\rangle$ to get a trainable model.

Leveraging the flexibility of basis encoding, Quark is capable of realizing modules used in classical deep models. Figure 4 illustrates two Quark modules (Conv, MaxPool) used in our experiments. Besides Convolution and MaxPooling, Quark can support more diverse operations given enough qubits. We include the demonstration of the Fully Connective and ReLU modules in Appendix A.4.

During training, we encode the entire data distribution through basis encoding so we can apply model forward simultaneously for all data samples. During inference, we encode each sample as a single state $|\hat{x}\rangle$. Prediction can be obtained by measurements over the output register after the model forward step $U_M|w^*\rangle|\hat{x}, 0\rangle|0\rangle = |w^*\rangle|\hat{x}, 0\rangle|f(w^*, \hat{x})\rangle$, where $w^*$ is the measured optimal weight.

### 4.2 AMPLITUDE AMPLIFICATION

To increase the probability of measuring the weights with the highest classification accuracy, we use Grover's algorithm to amplify the amplitude for any state $|w_i\rangle|x_j, y_j\rangle|f(w_i, x_j)\rangle \in \mathcal{S}$, thus the measuring probability of optimal weights would be amplified the most. In this section, we will formally define the unitary operators we need for Grover's update, followed by the complexity analysis of the optimization algorithm.

For conventional Grover's algorithm, we need two reflection unitaries, namely $U_{s^\top}, U_{\Psi_0}$, where $U_{s^\top}$ is the reflection against the non-solution sub-space and $U_{\Psi_0}$ is the reflection against the initial state. In our case, by defining solution state

$$|s\rangle = \sum_{|w_i\rangle|x_j, y_j\rangle|f(w_i, x_j)\rangle \in \mathcal{S}} \frac{1}{\sqrt{|\mathcal{S}|}} |w_i\rangle|x_j, y_j\rangle|f(w_i, x_j)\rangle$$

and initial state

$$\Psi_0 = \sum_{i,j} \frac{1}{\sqrt{2^{d_w}|\mathcal{D}|}} |w_i\rangle|x_j, y_j\rangle|f(w_i, x_j)\rangle$$

, the Grover operator can be easily built from $U_{\text{comb}} = U_{\Psi_0} U_{s^\top}$ where $U_{s^\top} = \mathbb{I} - 2|s\rangle\langle s|, U_{\Psi_0} = 2|\Psi_0\rangle\langle\Psi_0| - \mathbb{I}$.

Now that we have the Grover operator well-defined, we can formally do analysis on the algorithm in terms of model query complexity.

**Theorem 1** *By defining* $\alpha = \frac{\int_{w_i \in \mathcal{W}_\epsilon} J(w_i) d\delta}{\int_{w_i \in \mathcal{W}} J(w_i) d\delta}$ *as the probability to measure an $\epsilon$-optimal solution after the Grover's update, the model query complexity for getting an $\epsilon$-optimal solution $w_i \in \mathcal{W}_\epsilon$ on a balanced $C$-way classification task is $O(\frac{\sqrt{C}}{\alpha})$.*

$J(w_i)$ is the objective value (accuracy) for weights $w_i$, $\mathcal{W}$ is the complete weight space, and $\mathcal{W}_\epsilon$ is the $\epsilon$-optimal weight subspace. The proof of Theorem 1 is included in Appendix A.3.1, it follows from the analysis that we need $O(\sqrt{C})$ Grover iterations per measurement and $O(\frac{\int_{w_i \in \mathcal{W}} J(w_i) d\delta}{\int_{w_i \in \mathcal{W}_\epsilon} J(w_i) d\delta})$ measurements in expectation for sampling an $\epsilon$-optimal weight.

Notice that the complexity of our method does not depend on sample size $N$. As we are sampling solutions from an optimized distribution for a general non-convex problem, our method does depend on an $O(\frac{\int_{w_i \in \mathcal{W}} J(w_i) d\delta}{\int_{w_i \in \mathcal{W}_\epsilon} J(w_i) d\delta})$ term. However, to achieve an $\epsilon$-optimal solution on a general non-convex problem, the worst case scenario for VQC-based methods with gradient-based optimizers needs $O(\frac{\int_{w_i \in \mathcal{W}} d\delta}{\int_{w_i \in \mathcal{W}_\epsilon} d\delta})$ iterations to sample initial points lie within convex regions that contain $\epsilon$-optimal solutions. With per iteration gradient evaluation cost that is in the order of $O(N)$, this gives an overall complexity of $O(\frac{\int_{w_i \in \mathcal{W}} d\delta}{\int_{w_i \in \mathcal{W}_\epsilon} d\delta} N)$ for VQC-based methods. Since $O(\frac{\int_{w_i \in \mathcal{W}} J(w_i) d\delta}{\int_{w_i \in \mathcal{W}_\epsilon} J(w_i) d\delta}) < O(\frac{\int_{w_i \in \mathcal{W}} d\delta}{\int_{w_i \in \mathcal{W}_\epsilon} d\delta})$ and $\sqrt{C} \ll N$ in practice. Our method provides a speed-up for finding an $\epsilon$-optimal solution in the general non-convex setup.

In addition, as Quark does not require gradients calculation, it will not suffer from the notorious problem of barren plateau. For more idealistic case of convex problems, Garg et al. (2020) has given a proof of no quantum speed-ups can be obtained in this case with no exception of our method.

### 4.3 K-PARALLEL DATASETS (KPD)

A naive implementation can only achieve $p(w_i) \propto J(w_i)$, which requires more measurements for sampling $\epsilon$-optimal solution. To this end, we further extend our method to achieve $p(w_i) \propto J(w_i)^k$ through $K$-Parallel Dataset (KPD), as shown in Figure 1 when $k > 1$. The intuition is similar to simulated annealing, by updating weights distribution proportional to $J(w_i)^k$, the probability mass would gradually converge to the global optimal solutions.

We do so by concatenating the same dataset $K$ times for increasing solution states in the order of $K$ as:

$$U_M \left( |W\rangle \bigotimes_{k=0}^{K-1} (|D_k\rangle \otimes |O_k\rangle) \right) = \sum_i \frac{1}{2^{\frac{d_w}{2}}} |w_i\rangle \prod_{k=0}^{K-1} \left( \sum_j \frac{1}{\sqrt{|\mathcal{D}_k|}} |x_j, y_j\rangle |f(w_i, x_j)\rangle \right)$$

the solution set is now defined as:

$$\mathcal{S}_K : \{ |w_i\rangle \bigotimes_{k=0}^{K-1} |x_{j_k}, y_{j_k}\rangle |f(w_i, x_{j_k})\rangle \mid \bigwedge_{k=0}^{K-1} (y_{j_k} = f(w_i, x_{j_k})) \}$$

We then update $|s\rangle, |\Psi_0\rangle, U_{comb}$ accordingly as in Appendix A.2. Now the ratio of solution states between different weights can grow exponentially in terms of $K$.

**Theorem 2** *By defining* $\beta = \frac{\int_{w_i \in \mathcal{W}_\epsilon} d\delta}{\int_{w_i \in \mathcal{W}/\mathcal{W}_\epsilon} d\delta}$ *as the volume ratio between $\epsilon$-optimal weight subspace* $|\mathcal{W}_\epsilon|$ *and non-$\epsilon$-optimal weight subspace* $|\mathcal{W} - \mathcal{W}_\epsilon|$, *the model query complexity for getting an $\epsilon$-optimal solution* $w_i \in \mathcal{W}_\epsilon$ *on a balanced $C$-way classification task with $k$-parallel dataset is* $O((1 + \beta^{k-1}(\frac{1}{\alpha} - 1)^k) k C^{\frac{k}{2}})$.

The proof of Theorem 2 is included in Appendix A.3.2. Here, the trade-off is that the number of Grover iterations needed per measurement grows to $O\left(C^{\frac{k}{2}}\right)$. As we also need to apply the model forward for each dataset, the KPD Grover complexity in terms of model query complexity is now $O(kC^{\frac{k}{2}})$. On the other hand, as we are converging to the global solution, the number of measurements needed can be reduced from $O(\frac{1}{\alpha})$ to $O(1 + \beta^{k-1}(\frac{1}{\alpha} - 1)^k)$ as $\beta < \alpha$. Thus the optimal value for $k$ depends on the specification of $\alpha, \beta, C$.

**Theorem 3** *Given $\alpha, \beta, C$, the optimal value of $k$ is* $k^* = \lfloor \log_{\frac{\alpha}{\beta}} \frac{1}{\alpha} \rfloor + 1$ *if* $\frac{\alpha}{\beta} \geq m^{\frac{1}{m-1}} \sqrt{C}$ *where* $m = \lfloor \log_{\frac{\alpha}{\beta}} \frac{1}{\alpha} \rfloor + 1$ *else $k^* = 1$.*

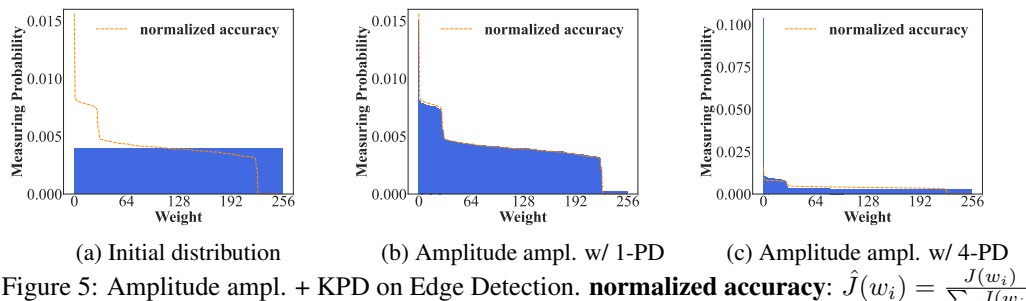

(a) Initial distribution      (b) Amplitude ampl. w/ 1-PD      (c) Amplitude ampl. w/ 4-PD

Figure 5: Amplitude ampl. + KPD on Edge Detection. **normalized accuracy**: $\hat{J}(w_i) = \frac{J(w_i)}{\sum_i J(w_i)}$.

Thus for cases where $m^{\frac{1}{m-1}}\sqrt{C} \leq \frac{\alpha}{\beta} \ll \frac{1}{\alpha}$, we have our optimal $k > 1$ which proves the effectiveness of KPD under certain circumstances. We include the proof of Theorem 3 in Appendix A.3.3.

## 5 EXPERIMENTS

In this section, we empirically verify the effectiveness of Quark on two tasks, namely Edge Detection and Tiny-MNIST. Algorithm 1 formally states the pipeline of Quark[5]. Instead of leveraging the approximated iteration number from theoretical derivations for Grover's update, we use a more precise but efficient estimation in practice to achieve a more accurate amplitude amplification effect. We include the detail of Grover iteration number estimation in Appendix A.5.1. Notice that most of the simulation results are obtained numerically since we have no access to large scale quantum devices that fit our setup. However, we do use Qiskit Aer (Anis & et al., 2021) to verify the reproducibility of our numerical simulation results on models that are applicable, results are included in Appendix A.6.

---

**Algorithm 1:** Quark's optimization pipeline.

---

**Input:** Data Oracle: $U_D$; Model Oracle: $U_M$; Objective Oracle: $U_L$; Number of parallel dataset: $k$; Measurement budget: $m$; Weights buffer: $\mathcal{B}$

**Output:** Optimized Model Weight: $w^*$

$g, U_{\text{comb}}, |\Psi_0\rangle = $ **Preprocessing**$(U_D, U_M, U_L, k)$ ;    `// get Grover iteration g,`
 `Grover operator` $U_{\text{comb}}$ `and initial state` $|\Psi_0\rangle$

**for** $i = 0; i < m; ++i$ **do**
   **for** $j = 0; j < g; ++j$ **do**
      $|\Psi_{j+1}\rangle = U_{\text{comb}}|\Psi_j\rangle$ ;                `// Grover update`
   $w_i = $ **Measure**$(|\Psi_g\rangle)$ ;            `// Weights measurement on` $R_W$
   $\mathcal{B}$.add$(w_i)$;

**return** $\arg\max_{w \in \mathcal{B}}($**Evaluate**$(w))$

---

### 5.1 EDGE DETECTION

For Edge Detection, our goal is to identify if a $3 \times 3$ binary matrix has 1) both vertical and horizontal lines 2) vertical lines only 3) horizontal lines only 4) no lines, where a line is defined by three consecutive '1's in a row or column. Thus the task is a 4-way classification task with 512 instances. We split the 512 samples into a training set with 400 randomly selected instances and a test set with the rest. We use a Quantum Convolutional Model that consists of several $1 \times 3$ convolution kernels with Maxpooling modules as described in Figure 4 for this task. Results are demonstrated in Figure 5 (normalized accuracy: $\hat{J}(w_i) = \frac{J(w_i)}{\sum_i J(w_i)}$), from which we can clearly see a linear relationship between model accuracy and optimized weights distribution using 1-PD. For 4-PD, we can observe that weights distribution further concentrates on the global optimal solutions.

Indeed, as we concatenate more training datasets, the probability mass will gradually converge to the optimal solutions, thus reducing the number of measurements we need significantly. We include the results for 2-PD and 3-PD in Appendix A.5.3.

---

[5]**Preprocessing**() and **Evaluate**() are included in Appendix A.5.1

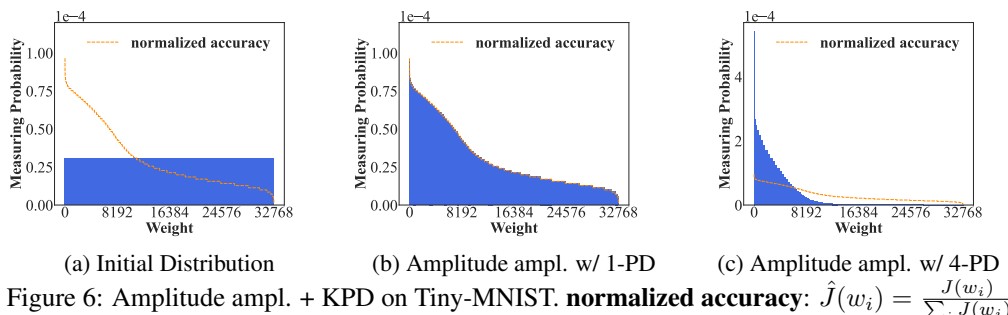

(a) Initial Distribution     (b) Amplitude ampl. w/ 1-PD     (c) Amplitude ampl. w/ 4-PD

Figure 6: Amplitude ampl. + KPD on Tiny-MNIST. **normalized accuracy**: $\hat{J}(w_i) = \frac{J(w_i)}{\sum_i J(w_i)}$.

## 5.2 TINY-MNIST

For Tiny-MNIST, we down-sample original $28 \times 28$ images from MNIST to $3 \times 3$ and remove duplicated samples. Due to device limitation, we only consider classes $1, 2, 7$, which makes this task a 3-way classification task. We use Weighted Mask modules (details are included in Appendix A.4) with MaxPooling modules to compose our model, which can achieve $86.08\%$ training accuracy and $82.61\%$ testing accuracy by the best model in the search space. Whereas a well-optimized classical single layer perceptron model can achieve $85\%$ in training accuracy and $80\%$ test accuracy.

Similarly, as shown in Figure 6 (normalized accuracy: $\hat{J}(w_i) = \frac{J(w_i)}{\sum_i J(w_i)}$), a linear relationship between model accuracy and optimized weights distribution can be observed using 1-PD, while 4-PD can further increase the probability for sampling optimal solutions. We also include the results of 2-PD and 3-PD on Tiny-MNIST in Appendix A.5.4.

Besides distribution evolution, we also statistically demonstrate the relationship between measured top-1 model's test accuracy and measurements budget using uniform random sampling, 1-PD, and 4-PD respectively in Figure 7. In order to achieve equally well-performed model, 4-PD only requires $\sim 30$ shots on Edge Detection task for a mean test accuracy $> 98\%$ while uniform random sampling needs $\sim 900$ ($\sim 30\times$ more) shots with much higher variance. Similar trend can be observed on Tiny-MNIST where 4-PD only requires $\sim 1000$ shots for a mean test accuracy $> 76.5\%$ while uniform random sampling needs $\sim 20000$ ($\sim 20\times$ more) shots with higher variance. We include the training ones in Appendix A.5.5.

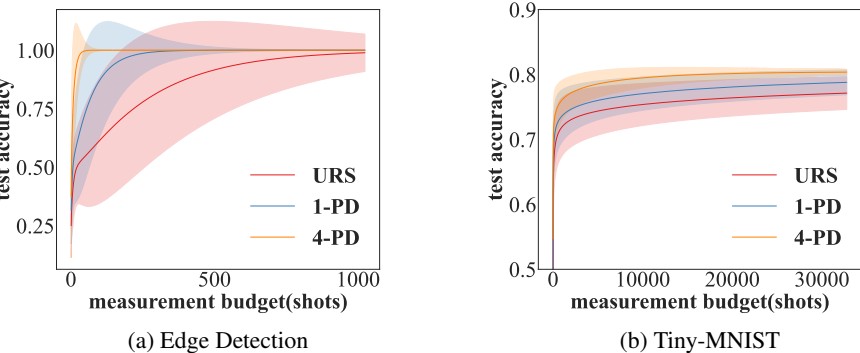

(a) Edge Detection        (b) Tiny-MNIST

Figure 7: The mean $\pm$ std of the test accuracy of the best discovered model with different measurement budget (in shots). URS shows the result where the weights are uniformly random sampled.

## 6 CONCLUSION

In this paper, we propose a new quantum learning framework Quark that does not involve gradients calculation and operates in a fully-quantum fashion. Acknowledging the notorious problem of barren plateaus from VQC based methods, Quark shed some lights on circumventing this phenomenon through a gradient-free optimization pipeline. Quark also enables a more general set of module design due to basis encoding, so that non-linear operations can be easily implemented. Theoretically, we present some evidences in terms of model query complexity for Quark to demonstrate trade-offs between VQC based methods and Quark. Empirically, we have verified the effectiveness of Quark through numerical simulations on Edge Detection and Tiny-MNIST.

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

# A APPENDIX

## A.1 GROVER'S ALGORITHM

A well-known algorithm for amplitude amplification is Grover's algorithm. The original Grover's algorithm is trying to search specific states that satisfy some properties which are called solution states. Instead of enumerating over all possible states to find the solution states that lie in the solution set $\mathcal{S}$, Grover's algorithm tries to amplify the solution states' amplitudes using two reflection unitary matrices $U_{s^\top}, U_{\Psi_0}$. Let $|\Psi_0\rangle$ denote the initial state of all qubits and $|s\rangle = \sum_{s_i \in \mathcal{S}} \frac{1}{\sqrt{|\mathcal{S}|}} |s_i\rangle$ represent the basis state spanned by all solution states. $U_{s^\top}$ and $U_{\Psi_0}$ are constructed as follows:

$$U_{s^\top} = \mathbb{I} - 2|s\rangle\langle s| \tag{2}$$

$$U_{\Psi_0} = 2|\Psi_0\rangle\langle\Psi_0| - \mathbb{I} \tag{3}$$

Geometrically, $U_{s^\top}$ is the reflection operator over $|s^\top\rangle = \sum_{s_i \notin \mathcal{S}} \frac{1}{\sqrt{|\{|0\rangle,|1\rangle\}^n \setminus \mathcal{S}|}} |s_i\rangle$, which is an state orthogonal to the solution space. Similarly, $U_{\Psi_0}$ is the reflection operator over $|\Psi_0\rangle$. Given $\theta = \arcsin\left(\langle\Psi_0|s^\top\rangle\right)$, $|\Psi_0\rangle$ can be expressed as:

$$|\Psi_0\rangle = \cos\theta|s^\top\rangle + \sin\theta|s\rangle$$

The combination of the two reflection unitary matrices $U_{\text{comb}} = U_{\Psi_0} U_{s^\top}$ is equivalent to a rotation of $2\theta$ on the plane spanned by $|s^\top\rangle$ and $|\Psi_0\rangle$. Therefore, applying the combination of the two reflection matrices $k$ times gives:

$$|\Psi_k\rangle = \left(\prod_{i=1}^{k} U_{\text{comb}}\right) |\Psi_0\rangle \tag{4}$$

$$= \cos\left((2k+1)\theta\right)|s^\top\rangle + \sin\left((2k+1)\theta\right)|s\rangle \tag{5}$$

As for most practical setups the solutions are always sparse and existed, we have $0 < \theta \ll \frac{\pi}{3}$. To maximize the amplitude for $|s\rangle$, $k$ should be in the order of $O(\frac{1}{\theta})$.

## A.2 DEFINITION

The $|s\rangle, |\Psi_0\rangle$ in KPD are updated as:

$$|s\rangle = \sum_{|w_i\rangle \otimes_{k=0}^{K-1} |x_{j_k}, y_{j_k}\rangle |f(w_i, x_{j_k})\rangle \in \mathcal{S}_K} \frac{1}{\sqrt{|\mathcal{S}_K|}} |w_i\rangle \bigotimes_{k=0}^{K-1} |x_{j_k}, y_{j_k}\rangle |f(w_i, x_{j_k})\rangle$$

and

$$\Psi_0 = \sum_{i, j_0, \cdots, j_{K-1}} \frac{1}{\sqrt{2^{d_w}|\mathcal{D}|^K}} |w_i\rangle \bigotimes_{k=0}^{K-1} |x_{j_k}, y_{j_k}\rangle |f(w_i, x_{j_k})\rangle$$

,again the Grover operator can be easily built from $U_{\text{comb}} = U_{s^\top} U_{\Psi_0}$ where $U_{s^\top} = \mathbb{I} - 2|s\rangle\langle s|, U_{\Psi_0} = 2|\Psi_0\rangle\langle\Psi_0| - \mathbb{I}$.

### A.3 PROOF

#### A.3.1 THEOREM 1

Given the probability for sampling a solution state is $p$, then the Grover iterations to achieve the maximum amplitude amplification effect is $\sqrt{\frac{1}{p}}$. As in our case, the probability for sampling a solution state is given by $\frac{\int_{w_i \in \mathcal{W}} J(w_i)d\delta}{\int_{w_i \in \mathcal{W}} d\delta}$, then suppose we are doing a balanced $C$-way classification task the Grover iterations we need is:

$$\sqrt{\frac{\int_{w_i \in \mathcal{W}} d\delta}{\int_{w_i \in \mathcal{W}} J(w_i)d\delta}} = \sqrt{\frac{1}{\mathbb{E}_{w_i \sim \mathcal{W}}[J(w_i)]}} \tag{6}$$

$$= \sqrt{\frac{1}{\frac{1}{C}}} \tag{7}$$

$$= \sqrt{C} \tag{8}$$

$(9) \rightarrow (10)$ is due to for a balanced $C$-way classification, we should expect the accuracy for a random model to be the same as a random guess $\frac{1}{C}$

As we have defined $\alpha = \frac{\int_{w_i \in \mathcal{W}_\epsilon} J(w_i)d\delta}{\int_{w_i \in \mathcal{W}} J(w_i)d\delta}$ as the probability to measure a $\epsilon$-optimal solution, thus in order to sample a $\epsilon$-optimal solution by measurements, it takes $O(\frac{1}{\alpha})$ trials. Which gives us an overall complexity of $O(\frac{\sqrt{C}}{\alpha})$

#### A.3.2 THEOREM 2

Given we are doing a balanced $C$-way classification task for $k$ parallel dataset, the probability for sampling a solution state is now $\frac{\int_{w_i \in \mathcal{W}} J(w_i)^k d\delta}{\int_{w_i \in \mathcal{W}} 1^k d\delta} = \mathbb{E}_{w_i \sim \mathcal{W}}[J(w_i)^k]$. Since the objective function we defined is non-negative $J(w_i) \geq 0$, thus we have $J(w_i)^k$ to be a convex function in $J(w_i)$. As expectation operator preserve convexity we have $\mathbb{E}[J(w_i)^k] \geq \mathbb{E}[J(w_i)]^k$, we have:

$$\sqrt{\frac{\int_{w_i \in \mathcal{W}} 1^k d\delta}{\int_{w_i \in \mathcal{W}} J(w_i)^k d\delta}} = \sqrt{\frac{1}{\mathbb{E}_{w_i \sim \mathcal{W}}[J(w_i)^k]}} \tag{9}$$

$$\leq \sqrt{\frac{1}{\mathbb{E}_{w_i \sim \mathcal{W}}[J(w_i)]^k}} \tag{10}$$

$$= \sqrt{C^k} \tag{11}$$

Thus the Grover iteration needed is upper bounded by $C^{\frac{k}{2}}$

For measurement, the expected iterations we need is $\frac{\int_{w_i \in \mathcal{W}} J(w_i)^k d\delta}{\int_{w_i \in \mathcal{W}_\epsilon} J(w_i)^k d\delta}$ which can be approximated as:

$$\frac{\int_{w_i \in \mathcal{W}} J(w_i)^k d\delta}{\int_{w_i \in \mathcal{W}_\epsilon} J(w_i)^k d\delta} = \frac{\int_{w_i \in \mathcal{W}_\epsilon} J(w_i)^k d\delta + \int_{w_i \in \mathcal{W}/\mathcal{W}_\epsilon} J(w_i)^k d\delta}{\int_{w_i \in \mathcal{W}_\epsilon} J(w_i)^k d\delta} \tag{12}$$

$$= 1 + \frac{\int_{w_i \in \mathcal{W}/\mathcal{W}_\epsilon} J(w_i)^k d\delta}{\int_{w_i \in \mathcal{W}_\epsilon} J(w_i)^k d\delta} \tag{13}$$

$$= 1 + \frac{\int_{w_i \in \mathcal{W}/\mathcal{W}_\epsilon} J(w_i)^k d\delta}{\int_{w_i \in \mathcal{W}_\epsilon} J(w_i)^k d\delta} \frac{\int_{w_i \in \mathcal{W}_\epsilon} 1^k d\delta}{\int_{w_i \in \mathcal{W}/\mathcal{W}_\epsilon} 1^k d\delta} \frac{\int_{w_i \in \mathcal{W}/\mathcal{W}_\epsilon} 1^k d\delta}{\int_{w_i \in \mathcal{W}_\epsilon} 1^k d\delta} \tag{14}$$

$$= 1 + \frac{\frac{\int_{w_i \in \mathcal{W}/\mathcal{W}_\epsilon} J(w_i)^k d\delta}{\int_{w_i \in \mathcal{W}/\mathcal{W}_\epsilon} 1^k d\delta}}{\frac{\int_{w_i \in \mathcal{W}_\epsilon} J(w_i)^k d\delta}{\int_{w_i \in \mathcal{W}_\epsilon} 1^k d\delta}} \frac{1}{\beta} \tag{15}$$

$$= 1 + \frac{\mathbb{E}_{w_i \in \mathcal{W}/\mathcal{W}_\epsilon}[J(w_i)^k]}{\mathbb{E}_{w_i \in \mathcal{W}_\epsilon}[J(w_i)^k]} \frac{1}{\beta} \tag{16}$$

$$\approx 1 + \frac{\mathbb{E}_{w_i \in \mathcal{W}/\mathcal{W}_\epsilon}[J(w_i)]^k}{\mathbb{E}_{w_i \in \mathcal{W}_\epsilon}[J(w_i)]^k} \frac{1}{\beta} \tag{17}$$

$$= 1 + (\frac{\mathbb{E}_{w_i \in \mathcal{W}/\mathcal{W}_\epsilon}[J(w_i)]}{\mathbb{E}_{w_i \in \mathcal{W}_\epsilon}[J(w_i)]})^k \frac{1}{\beta} \tag{18}$$

$$= 1 + (\frac{\frac{\int_{w_i \in \mathcal{W}/\mathcal{W}_\epsilon} J(w_i) d\delta}{\int_{w_i \in \mathcal{W}/\mathcal{W}_\epsilon} 1 d\delta}}{\frac{\int_{w_i \in \mathcal{W}_\epsilon} J(w_i) d\delta}{\int_{w_i \in \mathcal{W}_\epsilon} 1 d\delta}})^k \frac{1}{\beta} \tag{19}$$

$$= 1 + (\frac{\int_{w_i \in \mathcal{W}_\epsilon} 1 d\delta}{\int_{w_i \in \mathcal{W}/\mathcal{W}_\epsilon} 1 d\delta} \frac{\int_{w_i \in \mathcal{W}/\mathcal{W}_\epsilon} J(w_i) d\delta}{\int_{w_i \in \mathcal{W}_\epsilon} J(w_i) d\delta})^k \frac{1}{\beta} \tag{20}$$

$$= 1 + (\beta \frac{\int_{w_i \in \mathcal{W}} J(w_i) d\delta - \int_{w_i \in \mathcal{W}_\epsilon} J(w_i) d\delta}{\int_{w_i \in \mathcal{W}_\epsilon} J(w_i) d\delta})^k \frac{1}{\beta} \tag{21}$$

$$= 1 + (\beta(\frac{1}{\alpha} - 1))^k \frac{1}{\beta} \tag{22}$$

$$= 1 + \beta^{k-1}(\frac{1}{\alpha} - 1)^k \tag{23}$$

$(18) \rightarrow (19)$ is assuming $\mathbb{E}_{w_i \in \mathcal{W}/\mathcal{W}_\epsilon}[J(w_i)] > \text{Var}_{w_i \in \mathcal{W}/\mathcal{W}_\epsilon}[J(w_i)]^{\frac{1}{2}}$ which is usually the case in practice since $J(w_i)$ itself is upper bounded by $[0, 1 - O(\epsilon)]$ within the non-$\epsilon$ optimal region and $J(w_i)$ can be assumed to be near-uniformly distributed within $\mathcal{W}/\mathcal{W}_\epsilon$ across range $[0, 1 - O(\epsilon)]$, which is a general assumption. Thus $\frac{\mathbb{E}_{w_i \in \mathcal{W}/\mathcal{W}_\epsilon}[J(w_i)^k]}{\mathbb{E}_{w_i \in \mathcal{W}_\epsilon}[J(w_i)^k]}$ can be dominated by $\frac{\mathbb{E}_{w_i \in \mathcal{W}/\mathcal{W}_\epsilon}[J(w_i)]^k}{\mathbb{E}_{w_i \in \mathcal{W}_\epsilon}[J(w_i)^k]} \leq \frac{\mathbb{E}_{w_i \in \mathcal{W}/\mathcal{W}_\epsilon}[J(w_i)]^k}{\mathbb{E}_{w_i \in \mathcal{W}_\epsilon}[J(w_i)]^k}$.

### A.3.3 THEOREM 3

As the trade-off is between measurements needed versus Grover iterations per measurement, we can list their rate of change to make the comparison. For $k$-parallel dataset, Grover iterations is increased by $kC^{\frac{k-1}{2}} = \frac{kC^{\frac{k}{2}}}{C^{\frac{1}{2}}}$. For measurements needed, the exact ratio should be $\frac{\frac{1}{\alpha}}{1+\beta^{k-1}(\frac{1}{\alpha}-1)^k}$, however we can further approximate this ratio as $\frac{\frac{1}{\alpha}}{\frac{\beta^{k-1}}{\alpha^k}\frac{1}{\alpha}} = (\frac{\alpha}{\beta})^{k-1}$ given $\beta^{k-1}(\frac{1}{\alpha}-1)^k > 1$ which is equivalent to $k \leq \lfloor \log_{\frac{\alpha}{\beta}} \frac{1}{\alpha} \rfloor + 1$. Thus the optimal $k$ should satisfy $\frac{\alpha}{\beta} \geq k^{\frac{1}{k-1}} C^{\frac{1}{2}}$ as well as $k \leq \lfloor \log_{\frac{\alpha}{\beta}} \frac{1}{\alpha} \rfloor + 1$. As $k^{\frac{1}{k-1}} C^{\frac{1}{2}}, k \geq 2$ is monotonically decreasing with respect to $k$ thus as long as $\frac{\alpha}{\beta} \geq m^{\frac{1}{m-1}} C^{\frac{1}{2}}$ for $m = \lfloor \log_{\frac{\alpha}{\beta}} \frac{1}{\alpha} \rfloor + 1$ we could have our optimal $k = m = \lfloor \log_{\frac{\alpha}{\beta}} \frac{1}{\alpha} \rfloor + 1$, otherwise $k = 1$.

## A.4 MODULE DESIGN DETAILS

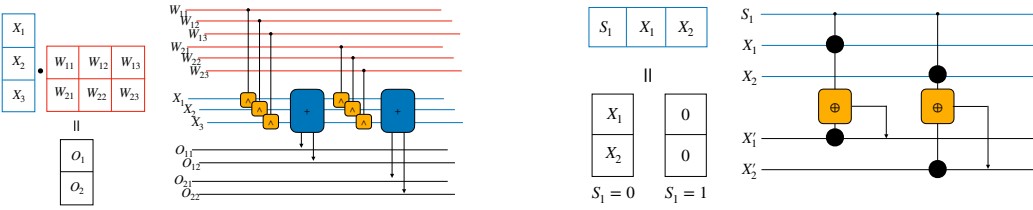

(a) Fully connected layer        (b) ReLU

Figure 8: Representing a fully connected layer and ReLU activation as quantum circuits in the Quark optimization framework. (a) Fully connective layer: for a $2 \times 3$ FC layer with weights $W_{ij}$ and input $X = [X_1, X_2, X_3]$, the output is given by $O_i = (W_{i1} \wedge X_1) + (W_{i2} \wedge X_2) + (W_{i3} \wedge X_3)$. (b) ReLU: for an input $X_i = [S, X_1, X_2]$ where $S$ is the sign qubit, the ReLU output is given by $X' = (X \oplus X) \vee (\neg S \wedge X)$.

In addition to the convolution and maxpooling layers shown in Figure 4a and Figure 4b, Quark can also incorporate other commonly used tensor algebra operators. Figure 8 demonstrates how to represent a fully connected and a ReLU layer as quantum circuits in Quark.

Due to the limitation of the number of qubits, our parameterization can be viewed as a binary model over a bounded weight space $\mathcal{W}$. In our experiments, we make a little modification to Quantum Learning modules introduced before. For Edge Detection task,

$$O_0 = (\bigvee_{i=0}^{2} (\bigwedge_{j=0}^{2} W_{0,j} \oplus X_{i,j})) \oplus W_{0,3} \tag{24}$$

$$O_1 = (\bigvee_{i=0}^{2} (\bigwedge_{j=0}^{2} W_{1,j} \oplus X_{j,i})) \oplus W_{1,3} \tag{25}$$

We use $|O_0 O_1\rangle = |00\rangle, |01\rangle, |10\rangle, |11\rangle$ to express the 4 different predictions respectively. For Tiny MNIST task,

$$O_k = (\bigvee_{i=0}^{2} (\bigvee_{j=0}^{2} W_{k,3i+j} X_{i,j})) \oplus W_{k,9} \quad k = 0,1 \tag{26}$$

We use $|O_0 O_1\rangle = |10\rangle, |11\rangle$ to express the prediction of Number 1 and use $|O_0 O_1\rangle = |01\rangle, |00\rangle$ to express the prediction of Number 2 and Number 7 respectively.

## A.5 EXPERIMENTS

### A.5.1 QUARK PROCESS

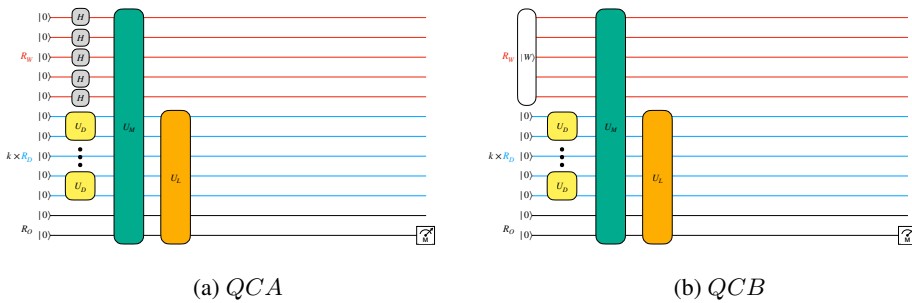

(a) $QCA$        (b) $QCB$

Figure 9: $QCA$ is the quantum circuit for calculating the average accuracy for different parameters, which is used in computing the number of Grover iterations as shown in Algorithm 2. $QCB$ shows the quantum circuit for computing the accuracy of a specific weight, which is used in Algorithm 3.

---

**Algorithm 2:** Get Grover Iterations Number

---

**Input:** Data Encoder Oracle: $U_D$; Model Forward Oracle: $U_M$; Objective Function Oracle: $U_L$; number of parallel dataset: $k$; Shots to estimate accuracy: $s$

**Output:** Grover Iterations Number: $g$

---

Initialize Obj $= 0$;
QC $=$ QCA$(U_D, U_M, U_L, k)$;    // Construct the quantum circuit shown in
 Figure 9(a)
**for** $i = 0; i < s; ++i$ **do**
 $\quad$ Obj $=$ Obj $+$ QC.measure$(R_O[-1])$
Obj $=$ Obj$/S$;
$\theta = \arcsin(\text{Obj})$;
$g = [\frac{m\pi + \frac{\pi}{2} - \theta}{2\theta}], \quad m \in \mathcal{N}$;
**return** $g$;

---

---

**Algorithm 3:** Evaluate

---

**Input:** Data Encoder Oracle: $U_D$; Model Forward Oracle: $U_M$; Loss Function Oracle: $U_L$; number of parallel dataset: $k$; Shots to estimate accuracy: $s$; Weight: $w$

**Output:** $J(w)$

---

Initialize $J = 0$;
QC $=$ QCB$(U_D, U_M, U_L, k, w)$; // Construct the quantum circuit shown in
 Figure 9(b)
**for** $i = 0; i < s; ++i$ **do**
 $\quad$ $J = J +$ QC.measure$(R_O[-1])$
$J = J/s$;
**return** $J$;

---

We use **Preprocessing**() to:

- Calculate number of Grover iterations we need (The procedure is illustrated in Figure 9(a) and Algorithm 2).

- Construct Grover Operator $U_{comb}$.

- Uniformly initialize $R_W$ with Hadamard gates.

- Initialize $k \times R_D$ with $U_D$s to encode k identical training dataset in parallel.

In Algorithm 2, we may find that $\theta$ is close to $\frac{\pi}{4}$, making $G$ extremely large or non-existent. However, we can introduce extra samples to dataset to resolve the issue. These auxiliary samples are identified by an additional qubit, that automatically being classified into non-solution space.

In function **Evaluate**(), we evaluate the objective value of a specific weight $w$. The procedure is illustrated in Figure 9(b) and Algorithm 3.

### A.5.2 TINY-MNIST EXPERIMENTAL SETUP

To construct Tiny-MNIST dataset, we select images with label 1, 2, 7 form the original MNIST training set and downsample them to 3x3 images with binarization applied to form $D_1$. As samples with same representations in $D_1$ cannot share different labels, we use majority voting to decide labels for duplicate samples within $D_1$ to form our final dataset. We apply same procedure for both training and testing datasets. This gives us a Tiny-MNIST dataset with 79 instances for training and 46 instances for testing. We use Tiny-MNIST for evaluating both classical methods and Quark. The settings of classical methods are shown in Table 1.

| Architecture | $9 \times 3$ FC + Softmax |
|---|---|
| Loss Function | CrossEntropyLosss |
| Optimizer | Adadelta |
| learning rate | 1.0 |
| batch size | 8 |
| epochs | 100 |
| seed | 0,1,2,3,4,5,6,7,8,9 |

Table 1: Basic Settings of Classical Method

### A.5.3 EDGE DETECTION RESULTS

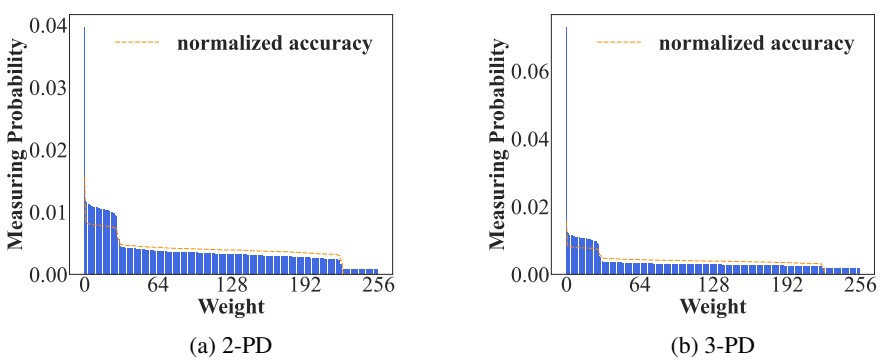

(a) 2-PD                    (b) 3-PD

Figure 10: The effect of amplitude amplification for Edge Detection.

### A.5.4 TINY-MNIST RESULTS

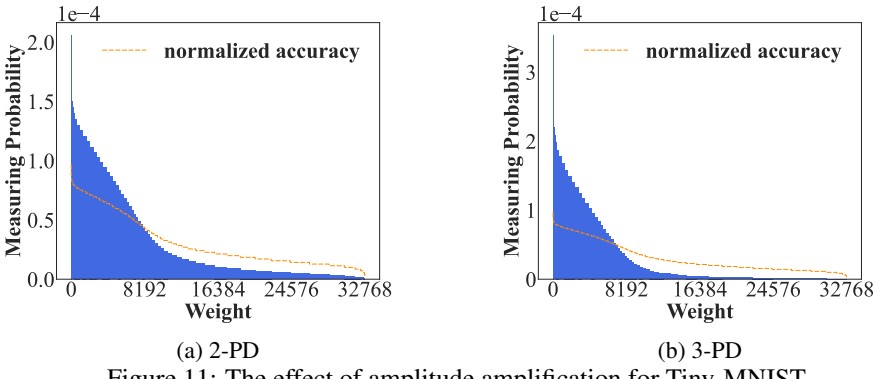

(a) 2-PD                    (b) 3-PD

Figure 11: The effect of amplitude amplification for Tiny-MNIST.

### A.5.5 SHOTS & TRAINING ACCURACY

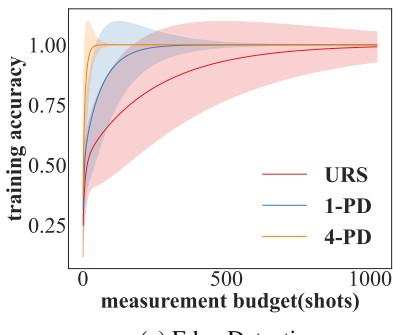 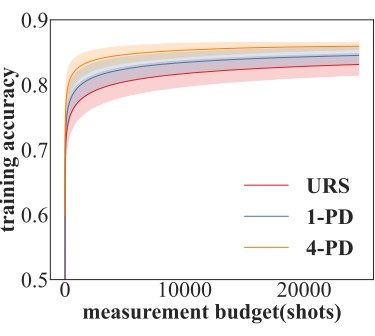

(a) Edge Detection                    (b) Tiny-MNIST

Figure 12: mean ± std of the best model's training accuracy measured within a given measurement budget.

### A.6 QUANTUM SIMULATION

To further verify our framework, we use Qiskit Aer (Anis & et al., 2021) to simulate the process of solving a simplified Edge Detection task with Quark. The goal is to identify whether a 3×3 binary matrix has horizontal lines. The task is a binary classification task with 512 instances. We randomly select 400 of them as the training set and the rest as the test set. The result of simulation is shown in Table 2. Each data point is acquired with 20 runs. The corresponding result of numerical simulation is shown in Table 3.

| shots | 1 | 2 | 4 | 8 | 16 | 32 | 64 | 128 |
|---|---|---|---|---|---|---|---|---|
| **train** | 59.14% | 67.9% | 68.36% | 75.33% | 85.04% | 85.79% | 91.93% | 94.14% |
| **std** | 12.21% | 10.34% | 9.05% | 11.26% | 9.27% | 9.01% | 6.18% | 4.30% |
| **test** | 58.17% | 65.04% | 65.54% | 75.63% | 84.69% | 84.55% | 90.18% | 92.41% |
| **std** | 13.54% | 10.35% | 9.60% | 11.32% | 10.09% | 10.59% | 7.33% | 5.77% |

Table 2: Relationships between training & test accuracy and shots. (Qiskit Simulation)

| shots | 1 | 2 | 4 | 8 | 16 | 32 | 64 | 128 |
|---|---|---|---|---|---|---|---|---|
| **train** | 60.74% | 66.20% | 71.52% | 77.49% | 83.82% | 89.59% | 93.93% | 96.88% |
| **std** | 10.72% | 10.26% | 10.96% | 11.12% | 10.09% | 7.87% | 5.53% | 3.86% |
| **test** | 60.03% | 64.99% | 70.11% | 76.18% | 82.83% | 88.90% | 93.22% | 95.96% |
| **std** | 11.17% | 11.07% | 11.97% | 12.28% | 11.23% | 8.72% | 6.11% | 4.95% |

Table 3: Relationships between training & test accuracy and shots. (Numerical Simulation)

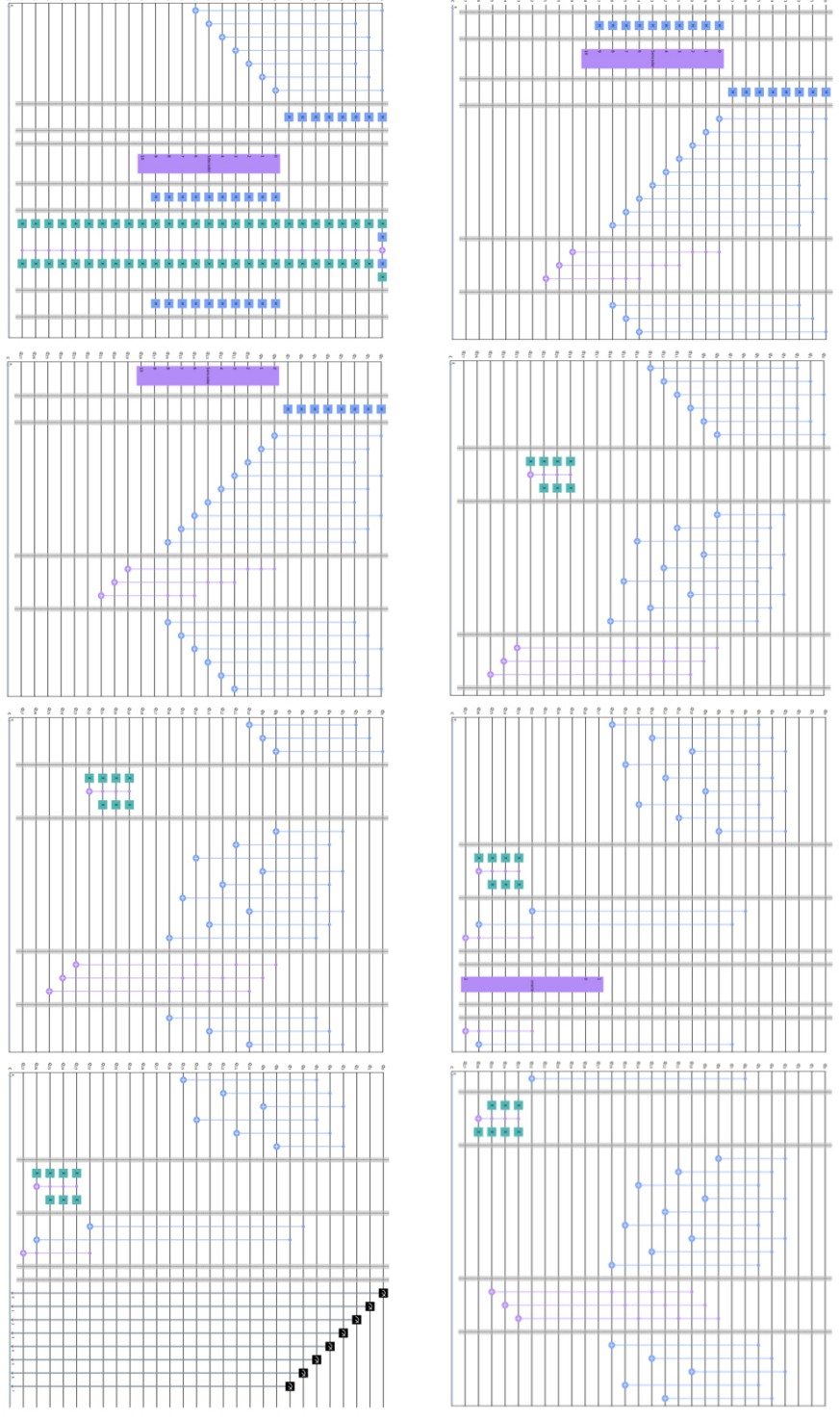

Figure 13: The quantum circuit for simplified edge detection.

