# OpenReview forum: "Quark: A Gradient-Free Quantum Learning Framework for Classification Tasks"
_ICLR.cc/2023/Conference — Submitted to ICLR 2023_

### Official Review · Reviewer_5Kqc · 2022-10-22

**Confidence:** 4
**Clarity, Quality, Novelty And Reproducibility:** See the weakness part and summary par…
**Correctness:** 3
**Technical Novelty And Significance:** 3
**Empirical Novelty And Significance:** 2
**Recommendation:** 3

**Strength And Weaknesses:**

Strength:
1. The paper is well organized with the whole approach demonstrated in a logical way. It is pleasant to read the paper.
2. Figures are properly used to illustrate points.
3. Classical simulation and Qiskit are used in the experiment to show the efficiency of the model.

Weakness:
1. My foremost concern is why quantum. The whole proposed method from my observation is using NEGATE, XOR, AND, OR on the qubits and all of them are classical operators. As in Figure 13, apart from the low resolution, I can barely see any other gates that are complex. It seems to me that all the framework, except for Grover, is in the field of Real numbers. Each qubit is used as a 0-1 indicator and the word “superposition” only occurred in Section 3.2 (Quantum basics). I just wonder why all the effort to design such a quantum paradigm with all the components classical (and the Grover is used to push the qubit to either 0 or 1 which is exactly the same as nonlinear unit in classical ml).

2. As for the first point in the contribution listed in section 1, the author said Quark is a gradient-free quantum learning framework. However, the Quantum model with classical optimization (QMCO) does not always use the classic gradient-based optimizers. For example, Qiskit uses a gradient-free method COBYLA to optimize VQE in the portfolio optimization tutorial [e]. There are many gradient-free optimizers that can be used in quantum models, so why not compare these methods?

3. In the second point in the contribution, the authors used the phrase “outperform gradient-based methods”. However, there is no evidence to show that the results of Quark are better than gradient based QML. I think the proper words can be “faster than gradient-based methods”.

4. In the third point in the contribution, the authors said “Quark can support complex ML models”. I think the authors ignored the qubit requirement of the proposed model, which is fatal to “complex ML models”. Using basis encoding itself requires lots of qubits as well as the qubits we need for all the parameters in the Weight and the Output and even repeat the data for k times. There are plenty of other QML approaches (e.g. QCNN [a], QRNN [b], QGAN [c], QLSTM [d]) and it is questionable for defining classification task as “complex” model.
5. How the model is trained is not clarified in the paper. How to tune the weights using the labels y_i is not shown in the algorithms. Besides, when it comes to the test stage, what should we do about the y_i in D is not clear. If the dimension is changed, then the training and testing circuits should be two different circuits?

6. In figure 8, the author proposed a FC layer and a ReLU circuits. The circuit for FC layer is wrong. W_1 and W_2 should act on X independently. However, according to the circuit, W_2 is applied to a state after W_1, which is different before W_1. As for 8(b), I don’t see the nonlinearity in the circuit since all the gates can be written in the unitary form and the number of qubits is not changed, maybe the author could further explain why this is nonlinear.

[a] Iris Cong, Soonwon Choi, and Mikhail D Lukin. 2019. Quantum convolutional neural networks. Nature Physics 15, 12 (2019), 1273–1278.
[b] Johannes Bausch. 2020. Recurrent quantum neural networks. Advances in neural information processing systems 33 (2020), 1368–1379.
[c] He-Liang Huang, Yuxuan Du, Ming Gong, Youwei Zhao, Yulin Wu, Chaoyue Wang, Shaowei Li, Futian Liang, Jin Lin, Yu Xu, Rui Yang, Tongliang Liu, Min- Hsiu Hsieh, Hui Deng, Hao Rong, Cheng-Zhi Peng, Chao-Yang Lu, Yu-Ao Chen, Dacheng Tao, Xiaobo Zhu, and Jian-Wei Pan. 2021. Experimental Quantum Generative Adversarial Networks for Image Generation. Physical Review Applied 16, 2 (2021).
[d] Samuel Yen-Chi Chen, Shinjae Yoo, and Yao-Lung L Fang. 2022. Quantum long short-term memory. In IEEE International Conference on Acoustics, Speech and Signal Processing. 8622–8626.
[e] https://qiskit.org/documentation/finance/tutorials/01_portfolio_optimization.html

**Summary Of The Paper:**

This paper proposed a novel gradient free quantum learning framework called QUARK, which is designed for classification tasks. Different numbers of qubits are assigned to Data, Weight and Output session, and the data are encoded with basis encoding method. Weights are applied to data through controlled CNOT gates and the weights are updated by Grover algorithm to amplify the amplitudes of weights. A few experiments are conducted on toy datasets and the results are reported.

**Summary Of The Review:**

This paper is well organized but the authors may not be enough familiar with quantum computing. Quantum machine learning does face the problems of barren plateau and nonlinearity, but the proposed model does not fulfill the contributions listed by the authors. The qubit requirement is huge with only classification tasks can be done, and the only advantage in time complexity is brought by the Grover search they employed in the model. The training and testing procedures are not clearly explained and several circuits in the paper are not correct. We seek for a more quantum-oriented solution to QML instead of using quantum as a shell.

---

> ### Author Response · Authors · 2022-11-16
> **Respond to reviewer 5Kqc Part (1/2)**
>
> We thank the reviewer for the reviews and answer the main concerns below.
>
> ### Q1. Why quantum
>
> Firstly, one of the key quantum advantages in our framework is to do classical operations in parallel for all possible weights and all samples in a dataset through a superposition over the entire weight space and sample space. Even though we don't deliberately use the word "superposition", we do entangle them in a superposed state, as introduced in section 4. In contrast, the amplitude/phase encoding + VQC-based method does not encounter such a benefit because the model forward is still applied sequentially for one single data point and one weight instantiation.
>
> Secondly, it's quantum parallelism and exponentially large representation space that bring quantum supremacy rather than having a complex domain. Considering successful classical deep models are all in the field of the real domain, we can thus conclude that operations in the real domain can already give us a good model.
>
> Thirdly, the reviewer may have misunderstood the functionality of Grover's algorithm in our framework. We use Grover's algorithm as an optimizer that pushes probability mass (amplitudes) towards optimal weights through amplitude amplification using Grover and KPD. In a future revision, we may include a more self-contained section to introduce the connection between Grover and our framework.
>
> ### Q2. Gradient-free optimizer
>
> Our title is a bit misleading here, we want to state that as far as we know, our Grover-based optimizer is the first gradient-free optimizer that is fully quantum for quantum ML models, which utilize the quantum parallelism to do global distributional optimization instead local optimization. And its potential effectiveness has been proven in sections 4.2 and 4.3. Meanwhile, its correctness has been verified through empirical simulations.
>
> One of the drawbacks of using classical optimizers, either gradient-dependent or gradient-free, is that they will induce frequent quantum classical interactions that potentially hinder quantum advantage.
>
> ### Q3. “outperform gradient-based methods.”
>
> We thank the reviewer for pointing out our misphrasing, we will change it in future revisions.

---

> > ### Author Response · Authors · 2022-11-16
> > **Respond to reviewer 5Kqc Part (2/2)**
> >
> > ### Q4. Complex ML models
> > We do acknowledge that our framework is not practical for near-term quantum devices. We want to state that our algorithm is not initially designed for near-term devices but rather aims for the quantum advantage when the quantum device scales.
> >
> > In addition, QCNN [a], QGAN [c] and QLSTM [d] all use amplitude/phase encoding, which lets the model can express nothing but a kernel method. Notice that even though QCNN uses a measurement-based model forward pipeline, it’s still a linear model in expectation which is not equivalent to the notion of non-linear models classically. QRNN  [b], on the other hand, also uses basis encoding to achieve non-linearity but a slightly different formulation, so it shares the same limitation (not near-term) as our work.
> >
> > ### Q5. Model training
> >
> > The training process is described in sections 4.2 and 4.3. We simultaneously evaluate all models on all data points using quantum parallelism (constructed superposed states) and push probability mass toward optimal weights using amplitude amplification (Grover’s algorithm). Thus our optimizer is different from conventional local optimization in that our Grover-based optimizer is a global optimizer in a fully quantum fashion.
> > We described how we do training and inference in section 4.1 (last paragraph). During inference, we can just drop the ground truth label circuit.
> >
> > ### Q6. Circuit for module
> >
> > We thank the reviewer for pointing out our typo in the figure of the FC layer’s circuit, which is a demo figure used to illustrate straightforward equivalency between classical models and models in our framework The FC circuit is not actually implemented in our experiments. We will modify this figure accordingly in our future revisions.
> >
> > Regarding non-linearity, amplitude/phase-based feature encoding methods are limited by the linear nature of unitaries. However, as we are using basis-based feature encoding, the amplitude is thus the entire data distribution, and the unitary constraint is linear on data distribution. But the non-linearity we state here is the function mapping of features that are encoded on the basis so that we can achieve any non-linearity just like classical operations using classical bits.

---

### Official Review · Reviewer_o76V · 2022-10-23

**Confidence:** 2
**Correctness:** 4
**Technical Novelty And Significance:** 3
**Empirical Novelty And Significance:** 3
**Recommendation:** 6

**Clarity, Quality, Novelty And Reproducibility:**

The paper is clearly written and it is of high quality. I believe the paper is novel but coming from a different domain I am not fully aware of related works.

**Strength And Weaknesses:**

The paper is well written and presented, and all derivations are technically sound. The paper sets a new direction towards unifying previous methods on quantum machine learning. The method is solid and worth a read, with a devoted supplementary material to get the good amount of content that cannot be fit onto the main document.

As a reader with interest in the topic but with no downstream-specific experience with hands-on on tools like Qiskit, I feel almost obliged to ask what is the main reason for the computational demand and why the experimental setting has to be reduced to very simple tasks such as edge detection in 3x3 matrices and a 3-class sub MNIST problem. The paper does not include any computational analysis neither by means of the used tool nor by the capacities of having real instrumentation. Why are the experiments limited to such a simple setting and what are the computational resources required for a simulation based on Qiskit? Knowing such information would allow potential researchers to find out which are the practical limitations in a real scenario where one can resort to simulations only.


**Summary Of The Paper:**

The paper proposes a quantum machine learning algorithm that can be trained without gradient descend, by applying the observation properties of quantum operators in a setting of superposition. The main idea is that a superposition is defined for the possible qubits describing the weights, datasets and outputs, with a function that produces the outputs according to the inputs and weights. Using Grover’s algorithm, the amplitudes for the probability of observing certain weights are amplified to maximize the training accuracy. Using many datasets in parallel and by simulating quantum operators, the method can learn different distributions of weights according to observations, setting a new direction towards a full end-to-end ``training” of quantum machine learning networks with new building blocks.

**Summary Of The Review:**

The paper sets a step towards full quantum machine learning systems, by a carefully designed machinery to allow defining a superposition of quantum weights, observables and outputs, and with an amplifier module to drive the probabilities of the weights to move towards maximizing the task-specific accuracy.

While the paper is technically novel, I miss some complexity analysis as well as a brief analysis of where do the authors think the main computational burden for the simulation comes from.

---

> ### Author Response · Authors · 2022-11-16
> **Respond to reviewer o76V**
>
> We thank the reviewer for the reviews and answer the main concerns below.
>
> ### Q1. Computational demand
>
> For a Qiskit-based simulator, with $n$ qubits, the simulated Unitary matrix can be of size $2^n \times 2^n$, which can lead to an Out-Of-Memory issue on a classical computer even if $ $n$$ is relatively small, like 30. So we can only resort to such simplified settings to demonstrate the correctness of our proposed framework.
>
> As our main contributions lie in algorithm design rather than a practical solution on existing near-term quantum devices, we do think our framework can potentially bring quantum advantage in the field of machine learning when quantum devices scale in the future.
>
> The problem with existing near-term orientated algorithms is that even though they can tackle much more complicated inputs, their success can be hindered by:
> 1. Frequent classical quantum interactions due to a classical optimizer in the loop
> 2. Limited model expressiveness due to amplitude/phase encoding of input features even if quantum device scales

---

### Official Review · Reviewer_YQCZ · 2022-10-24

**Confidence:** 4
**Correctness:** 2
**Technical Novelty And Significance:** 1
**Empirical Novelty And Significance:** 2
**Recommendation:** 1

**Clarity, Quality, Novelty And Reproducibility:**

The novelty of the approach is very limited - it relies on well-known ability to speed up search for good solutions (as provided by an arbitrary labeling function) by a polynomial factor using Grover's search / amplitude amplification. Due to the space and time complexity inherent to the method (see below), its significance for quantum machine learning in the near/medium term is also very limited.

It is well-known that if input is encoded as a sufficiently-long binary string (e.g. one qubit per classical bit), then unitary computation is capable of performing arbitrary logical functions, same as classical computer, and can thus compute convolutions, pooling, activation functions, etc. The problem is that this requires impractically large number of qubits - e.g., the authors' mention using $d_w$ qubits to encode the model, where $d_w$ is the total number of bits in all the trainable weights of the model.

The computational complexity analysis section mentions that the complexity does not depend on the training set size. Yet, consider number of samples that is below $N$, the VC-dimension of the model (for large networks, that could be a very large number of samples, larger than any practical dataset). Then, the model is capable of arbitrary labeling any training set of size $n \leq N$, indicating that the optimal (correct) labeling is one among $2^n$ possible labelings. In absence of any contrary analysis by the authors, one can assume that the volume of weights that lead to the optimal solution is exponentially (in $n$) small in the space of all possible weights - and the method has complexity growing with the inverse (in Thm 1, or the power of the inverse, in Thm 2) of that relative volume, i.e., complexity grows quite fast with $n$.

**Strength And Weaknesses:**

Strengths:
- having empirical evaluation of the quantum approach is a plus

Weaknesses:
- the basis encoding used in the method requires O(#features) qubits just to store the dataset, and additional O(#trainable parameters) qubits to store the model during computation. This severely limits the usefulness of the approach in a near and medium term.
- complexity analysis is expressed not in terms of input characteristics such as #samples, #features but instead (Thm 1) is is said to be (roughly) inversely proportional to the ratio of expected value of the objective function for optimal solutions to the expected value of the objective function for all possible solutions (all possible weights). This is smaller than the volume of optimal solutions among all possible solutions (authors' also mention that in pg.7). The problem is that for most reasonable models and classification problems, this volume is very small (otherwise, we would use random sampling to find a good model), yet authors do not attempt to offer any quantification of how small it is.
- the datasets used in the empirical evaluation are extremely small (9 features). Is this related to basis encoding being used?

**Summary Of The Paper:**

The paper presents an approach to training machine learning models using quantum computing. It is based on the well-known Grover's search / amplitude amplification approach.

**Summary Of The Review:**

The approach uses a well-known approach to speed-up any search problem via Grover's approach. The practicality of the approach for near/medium term quantum devices is very low. The computational complexity of the approach is not thoroughly assessed.

---

> ### Author Response · Authors · 2022-11-16
> **Respond to reviewer YQCZ Part (1/2)**
>
> We thank the reviewer for the reviews and answer the main concerns below.
>
> ### Q1. Basis encoding
>
> We do acknowledge that our framework is not practical for near-term quantum devices. We want to state that our algorithm is not initially designed for near-term devices but aims for the quantum advantage when the quantum device scales.
>
> We do think the quantum advantage in existing QML frameworks can be hindered by:
> 1. Frequent classical quantum interactions due to a classical optimizer in the loop
> 2. Limited model expressiveness due to amplitude/phase encoding of input features even if quantum device scales
>
> These bottlenecks still exist even when the quantum device scales. Thus, even though our framework is not practical on current quantum devices with limited qubits, we aim to solve the bottlenecks mentioned above on future scalable quantum devices.
>
> ### Q2. Complexity analysis
>
> Considering the worst-case scenario of general non-convex problems where gradient-based methods need to sample initial points over the entire weight space to find global optimal weights, which requires $O(\frac{\int_{w_i \in\mathcal{W}}d\delta}{\int_{w_i \in\mathcal{W_\epsilon}}d\delta})$ compared with our case $O(\frac{\int_{w_i \in\mathcal{W}}J(w_i)d\delta}{\int_{w_i \in\mathcal{W_\epsilon}}J(w_i)d\delta})$ that gives smaller iterations.
> In addition, naive implementation can indeed lead to large $O(\frac{\int_{w_i \in\mathcal{W}} J(w_i)d\delta}{\int_{w_i \in\mathcal{W_\epsilon}}J(w_i)d\delta})$ that grows exponentially to model size (which is correlated with the VC Dimension the reviewer mentioned). However, we do propose KPD to mitigate this issue by reducing $O(\frac{\int_{w_i \in\mathcal{W}}J(w_i)d\delta}{\int_{w_i \in\mathcal{W_\epsilon}}J(w_i)d\delta})$ to $ \frac{\int_{w_i \in\mathcal{W}}J(w_i)^k d\delta}{\int_{w_i \in\mathcal{W_\epsilon}}J(w_i)^k d\delta}$, so that as $k$ increases the probability mass also converge to optimal weights.

---

> > ### Author Response · Authors · 2022-11-16
> > **Respond to reviewer YQCZ Part (2/2)**
> >
> > ### Q3. Dataset
> >
> > Yes, we regard this as a trade-off between model expressiveness versus feature size. Suppose we are using amplitude/phase encoding for features, we do benefit from the exponentially large feature space but suffer from the model being kernel methods only, which is not comparable to classical deep learning models. Using basis encoding for features, on the other hand, enables quantum models to have more expressiveness and escape the kernel regime but at the cost of feature size.
> >
> > We do want to restate that our framework is not targeting near-term device but bring quantum advantage when quantum device scales, which cannot be achieved by amplitude/phase-based feature encoding methods whose expressiveness is limited even if we have more qubits.

---

### Official Review · Reviewer_Wci6 · 2022-10-26

**Confidence:** 5
**Clarity, Quality, Novelty And Reproducibility:** The paper is clear, the techniques ar…
**Correctness:** 3
**Technical Novelty And Significance:** 2
**Empirical Novelty And Significance:** 1
**Recommendation:** 3

**Strength And Weaknesses:**

+ new framework for quantum neural networks that avoids barren plateau
- using amplitude estimation within a quantum neural network makes it not really near-term
- there are many different ways of avoiding barren plateau that also keep the possibility of implementing the neural networks

**Summary Of The Paper:**

The authors present a gradient-free framework for quantum deep learning using known techniques for the optimization, in particular Grover's algorithm. The techniques are not really new and the scheme is not at all near-term as is shown from the lack of experiments.

**Summary Of The Review:**

Interesting idea but with many drawbacks and not enough novelty in the techniques

---

> ### Author Response · Authors · 2022-11-16
> **Respond to reviewer Wci6**
>
> We thank the reviewer for the reviews and answer the main concerns below.
> ### Q1. Using amplitude estimation within a quantum neural network makes it not really near-term
>
> We don’t actually use amplitude estimation in our pipeline but amplitude amplification as a global distributional optimizer to push probability mass toward optimal weights. However, we do acknowledge that our framework is not practical for near-term quantum devices. We want to state that our algorithm is not initially designed for near-term devices but rather aims for the quantum advantage when the quantum device scales.
>
> We do think the quantum advantage in existing QML frameworks can be hindered by:
>  1. Frequent classical quantum interactions due to a classical optimizer in the loop
>  2. Limited model expressiveness due to amplitude/phase encoding of input features, even if quantum device scales
>
> These bottlenecks still exist even with quantum device scales. Thus, even though our framework is not practical on current quantum devices with limited qubits, we aim to solve the bottlenecks mentioned above on future scalable quantum devices.
>
> ### Q2. There are many different ways of avoiding barren plateau that also keep the possibility of implementing the neural networks
>
> Our title is a bit misleading here. We want to state that as far as we know, our Grover-based optimizer is the first gradient-free optimizer that is fully quantum for quantum ML models, which utilizes quantum parallelism to do global distributional optimization instead of local optimization.
>
> Existing methods, either gradient-dependent or gradient-free optimizations for QML, mainly rely on a classical optimizer that can induce frequent quantum classical interactions.

---

### Decision · Program_Chairs · 2023-01-20

**Decision:**

Reject

**Justification For Why Not Higher Score:**

Clear from metareview and scores from reviewers.

**Justification For Why Not Lower Score:**

N/A

**Metareview: Summary, Strengths And Weaknesses:**

The paper presents a gradient-free framework for quantum deep learning based on Grover's algorithm. The reviewers agree that the approach taken in the paper is not particularly new. There is also a lack of discussion of prior work. This is more minor in my view but some reviewers also pointed out the experimental evaluation could be more exhaustive (using larger datasets).

I note that there was very little to no discussion between the reviewers due to a large agreement the paper is not ready for publication. I myself largely agree with the reviewers. I'm therefore not able to recommend this paper for acceptance. I encourage the authors to address the complaints of the reviewers, and perhaps resubmit to a more specialized conference or journal in quantum computing.

**Summary Of Ac-Reviewer Meeting:**

No meeting.